# NF-κB Decoy Oligodeoxynucleotide-Loaded Poly Lactic-co-glycolic Acid Nanospheres Facilitate Socket Healing in Orthodontic Tooth Movement

**DOI:** 10.3390/ijms25105223

**Published:** 2024-05-10

**Authors:** Albert chun-shuo Huang, Yuji Ishida, Kasumi Hatano-sato, Shuji Oishi, Jun Hosomichi, Risa Usumi-fujita, Hiroyuki Yamaguchi, Hiroyuki Tsujimoto, Aiko Sasai, Ayaka Ochi, Takashi Ono

**Affiliations:** 1Department of Orthodontic Science, Graduate School of Medical and Dental Sciences, Tokyo Medical and Dental University (TMDU), Tokyo 113-8549, Japan; albertpurinhaung@hotmail.com (A.c.-s.H.); k.hatano.orts@gmail.com (K.H.-s.); s.oishi.orts@gmail.com (S.O.); hosomichi.orts@tmd.ac.jp (J.H.); r.usumi.orts@tmd.ac.jp (R.U.-f.); t.ono.orts@tmd.ac.jp (T.O.); 2Department of Pediatrics, McGovern Medical School, The University of Texas Health Science Center at Houston, Houston, TX 77030, USA; hiroyuki.yamaguchi@uth.tmc.edu; 3Pharmaceutical/Beauty Science Research Center, Material Business Division, Hosokawa Micron Corporation, Osaka 573-1132, Japan; hytsujimoto@hmc.hosokawa.com (H.T.); ayasutake@hmc.hosokawa.com (A.S.); aochi@hmc.hosokawa.com (A.O.)

**Keywords:** alveolar bone loss, decoy oligodeoxynucleotides, nanospheres, nuclear factor-kappa B, orthodontic tooth movement

## Abstract

Orthodontic space closure following tooth extraction is often hindered by alveolar bone deficiency. This study investigates the therapeutic use of nuclear factor-kappa B (NF-κB) decoy oligodeoxynucleotides loaded with polylactic-co-glycolic acid nanospheres (PLGA-NfDs) to mitigate alveolar bone loss during orthodontic tooth movement (OTM) following the bilateral extraction of maxillary first molars in a controlled experiment involving forty rats of OTM model with ethics approved. The decreased tendency of the OTM distance and inclination angle with increased bone volume and improved trabecular bone structure indicated minimized alveolar bone destruction. Reverse transcription-quantitative polymerase chain reaction and histomorphometric analysis demonstrated the suppression of inflammation and bone resorption by downregulating the expression of tartrate-resistant acid phosphatase, tumor necrosis factor-α, interleukin-1β, cathepsin K, NF-κB p65, and receptor activator of NF-κB ligand while provoking periodontal regeneration by upregulating the expression of alkaline phosphatase, transforming growth factor-β1, osteopontin, and fibroblast growth factor-2. Importantly, relative gene expression over the maxillary second molar compression side in proximity to the alveolus highlighted the pharmacological effect of intra-socket PLGA-NfD administration, as evidenced by elevated osteocalcin expression, indicative of enhanced osteocytogenesis. These findings emphasize that locally administered PLGA-NfD serves as an effective inflammatory suppressor and yields periodontal regenerative responses following tooth extraction.

## 1. Introduction

Tooth extraction is commonly performed to prevent dental crowding and improve the aesthetic outcome of patients undergoing orthodontic treatment. However, alveolar bone resorption and severe bone loss due to excessive inflammation often occur after tooth extraction, which can consequently allow adverse adjacent tooth movement [1]. Notably, alternative periodontal regenerative surgery may help prevent bone degradation resulting from tooth movement into intra-bony defects caused by early inflammatory resorption. Periodontal ligament (PDL) remodeling and surrounding apparatus have been reviewed and analyzed, indicating that they were triggered by tooth extraction [2]. Orthodontic tooth movement (OTM) into areas of alveolar abnormalities, such as extraction sockets, can cause a gingival recession, periodontal pocket deepening, root resorption, and pulp vitality loss [3,4]. Moreover, OTM is a primary factor contributing to resorptive remodeling of the alveolar ridge [5,6], reduction in alveolar bone volume, unfavorable architecture of the surrounding alveolar bone, and undesirably long healing time from tooth extraction. These phenomena can compromise treatment outcomes, particularly during the early inflammatory-healing stage post-extraction. This highlights the importance of addressing these challenges to optimize orthodontic and periodontal health [7,8]. In summary, these adverse effects may lead to problematic space closure during orthodontic treatment [9].

Double-stranded transcription factor decoy oligodeoxynucleotides (ODNs) are therapeutic candidates capable of targeting and neutralizing key transcription factors implicated in the pathogenesis of inflammatory bone diseases [10]. Biological modulators that suppress osteoclast activity can be employed to solve the above-mentioned issues, representing novel supplementary orthodontic treatment approaches for alveolar bone deformities [11,12]. For example, suppressors of nuclear factor-kappa B (NF-κB) have been extensively investigated in the context of osteoclastic inhibition, given their pivotal role in controlling the survival, activation, and development of innate immune and inflammatory T cells during inflammatory bone disorders [13,14]. A previous in vitro study showed that NF-κB-decoy ODNs can inhibit osteoclast development and activation and that they can attenuate bone resorption as well as femur and tibia destruction when injected into rats with osteoporosis [15]. Moreover, local delivery of NF-κB-targeting ODNs has also proven effective in treating rheumatoid arthritis, periodontal bone defects, and extraction alveolar socket repair [16,17].

Increased ODN cellular uptake efficiency is considered a critical factor associated with the inhibitory effect on the DNA binding activity and downstream factors of NF-κB [18]. Using polylactic-co-glycolic acid (PLGA) nanospheres (NS) as a delivery system for ODNs against NF-κB showed that it allowed sustained ODN release with high intracellular uptake and substantial inhibition of NF-κB transcriptional activity [19]. Moreover, the long-term effects of decoy ODNs are determined by the stability and efficiency with which they are transported to the target tissues and absorbed by target cells. We previously reported that administration of NF-κB-decoy ODN-loaded PLGA NS (PLGA-NfDs) suppresses the inflammatory response and promotes bone regeneration over the alveolar ridge following tooth extraction in rats [17]. Given the unresolved issue of the resorptive ridge, we explored the therapeutic effect of PLGA-NfDs on the OTM biological response following tooth extraction, focusing on periodontal regeneration. We aimed to examine the therapeutic effects of local administration of PLGA-NfDs in bone remodeling events, in particular on periodontal regeneration in tooth movement and changes in the alveolar bone structure following tooth extraction, using an OTM rat model.

## 2. Results

### 2.1. 3D Micro-Computed Tomography (CT) Analysis

3D micro-CT analysis demonstrated a decrease in the OTM distance and inclination angle in PLGA-NfD-treated samples, without a significant difference among scrambled decoy ODN (ScD), NF-κB decoy ODN (NfD), and PLGA scrambled decoy ODN (PLGA-ScD) groups (Figure 1D,E). Representative 3D images of the VOI1 and VOI2 in all groups are shown in Appendix A. Compared with NfDs and PLGA-ScDs, PLGA-NfDs demonstrated a significant increase in BV/TV, bone mineral density (BMD), and Tb.N on D7 and D28 in both VOI1 and VOI2 regions, whereas Tb.Sp decreased significantly. No significant difference was observed between the effects of NfDs and those of ScD on D7 and D28 (Figure 2).

### 2.2. Tartrate-Resistant Acid Phosphatase (TRAP) and Alkaline Phosphatase (ALP) Staining

On D7, PLGA-NfD-treated samples showed a significant decrease in the number of TRAP-positive osteoclast cells/mm^2^ compared with that in the other three groups, whereas a larger ALP-positive stained area was observed in PLGA-NfD samples. NfDs exhibited a similar tendency, with less enhanced histological reactions. Significant results were obtained with TRAP and ALP staining of PLGA-NfD samples on D28 as compared with the other three groups (Figure 3A,D; Appendix A).

### 2.3. Biochemical Analysis

Relative expression of *Tnf* and *Il1b* decreased significantly in the PLGA-NfD group compared with that in the ScD group on D7 and D28, whereas a significant difference was observed in *Tnf* expression in the NfD group compared with that in the ScD group (Figure 4A,B). Regarding osteoclastogenesis-related genes, the expression of *Ctsk*, *Tnfsf11*, and *Tnfrsf11b* significantly decreased in the PLGA-NfD group on D7 and D28. The RANKL/OPG ratio in the PLGA-NfD group was significantly lower than that in the other groups on D28, whereas no significant difference was observed in D7 (Figure 4C and Figure 5C,F; Appendix A). Regarding osteogenesis-related genes, the expression of *Tgfb1*, *Spp1*, and *Bglap* in the PLGA-NfD group increased significantly on both D7 and D28, with the same result being observed for *Fgf2* expression related to PDL regeneration (Figure 4D–G).

### 2.4. Immunohistochemistry and Immunofluorescence Staining

Immunohistochemical analyses revealed that positive staining for specific markers was mainly observed in the cervical region of the periodontium and crestal alveolus on the M2 mesial side. The PLGA-NfD group demonstrated a significant reduction in the immuno-positive stained ratio of tumor necrosis factor (TNF)-α and interleukin (IL)-1β on D7 and D28 (Figure 3B,C; Appendix A). In contrast, bone formation was promoted, as evidenced by the significant increase in immuno-positive stained ratios of transforming growth factor (TGF)-β1 and osteopontin (OPN) in the PLGA-NfD group on D7 and D28 (Figure 3E,F; Appendix A). Immunofluorescence analysis further showed that NF-κB p65 and cathepsin K (CTSK) expression decreased in the PLGA-NfD group on D28 relative to the other three groups, whereas OPN and osteocalcin (OCN) levels showed an increasing tendency with a significant difference compared with that in the ScD group (Figure 5A,B,D,E).

## 3. Discussion

Space closure is frequently performed during orthodontic treatment after tooth extraction [20]. The longer the time after tooth extraction, the more ominous the sequelae; therefore, orthodontists should be cautious about the aftereffects that might appear in the residual alveolar ridge [21]. Reconstruction of the appropriate alveolar bone mass along with periodontal regeneration is crucial in facilitating OTM and tooth translation during extraction space closure, in addition to the proper orthodontic alignment of neighboring teeth at the extraction site [22,23]. Our previous study demonstrated the healing effect of PLGA-NfDs on the first-stage tooth-extraction socket in vivo [17]. In the current study, we explored the effect of the PLGA-NfD on the regulation of alveolar bone resorption and periodontal regeneration during OTM in rats subjected to orthodontic forces.

NF-κB expression rapidly increases in association with p65 in OTM, including osteoclastogenesis [24]. NF-κB decoy ODNs can suppress genetic activation by preventing essential transcriptional factors, such as the NF-κB p65—IκB complex, from binding to the promoter regions of target genes [25,26,27]. To evaluate the effect of PLGA-NfDs on the nuclear translocation of p65 [28], we evaluated NF-κB p65 expression and observed reduced levels in accordance with the CTSK expression profile, indicating a substantial impact of PLGA-NfDs following OTM in the extraction socket. Conventional pharmacological inhibitors of the NF-κB pathway, including non-steroidal anti-inflammatory drugs, are primarily non-specific inhibitors of NF-κB [29,30]. Thus, such drugs might block OTM by reducing the inflammatory or bone resorption processes [31]. Nonetheless, in the present study, PLGA-NfDs demonstrated a tendency to decrease OTM distance and inclination angle, although the differences were not statistically significant. In addition, increased BV/TV, BMD, and Tb.N, along with decreased Tb.Sp, confirmed denser trabecular bone with improved quality. These findings suggest that PLGA-NfD treatment rescued bone destruction as compared with the therapeutic approaches, implying reduced detrimental effects inherent to these therapeutic modalities while enhancing OTM. These changes may be associated with the ability to restore NF-κB transcriptional activity back to normal levels, which differs from the action mechanism of other NF-κB inhibitors [32].

Although OTM has been described as an aseptic inflammatory process, it has also been reported as an exaggerated form of normal physiological turnover combined with foci of tissue repair, leading to cementum and alveolar bone remodeling [33]. The inflammatory response is essential for osteoclast recruitment and differentiation and plays a vital role in alveolar bone and PDL remodeling during OTM following tooth extraction. This inflammatory response alters the PDL microenvironment, resulting in the release of proinflammatory cytokines, such as TNF-α and IL-1β, which are involved in bone remodeling signaling pathways and can trigger acute and chronic inflammation [34,35]. In the present study, PLGA-NfDs demonstrated decreased TNF-α and IL-1β expression on the compression side of the M2 alveolus, exhibiting the adjacent pharmacological influence of intra-socket PLGA-NfD administration over M2 OTM. Inhibition of NF-κB p65 activity with decoy ODNs may suppress the stimulatory effects of RANK/RANKL signaling in osteoblasts/osteoclasts, upregulating OPG expression in mesenchymal stem cells/preosteoblasts via an independent pathway. The consequent reduction in the RANKL/OPG ratio, followed by a decrease in RANKL and OPG levels after PLGA-NfD administration, may result from the suppression of osteoclastogenesis [36].

Preservation of PDL cells within the bony defect and extraction socket is deemed critical for periodontal regeneration during OTM. Previous studies have confirmed that PDL fibroblasts play a crucial role in OTM via NF-κB activation, whereas PDL destruction is more likely to occur if the inflammatory reaction is not well-regulated prior to bone resorption. In the present study, NF-κB inhibition by PLGA-NfDs blocked the OTM, as evidenced by the significant reduction in osteoclastogenesis, high BV/TV, and reduced RANKL expression, whereas increased TGF-β1 and FGF-2 expression was observed in the compression side of PDL compared with those in other treatment groups, along with an increased expression of ALP and OPN. In response to the mechanical stress applied to the periodontium, osteocytes are considered crucial factors in osteogenesis, and the osteoblast markers RUNX2, OSX, and OCN are also expressed in the PDL during OTM [37,38,39]. In the present study, treatment with PLGA-NfDs led to increased OPN and OCN expression on D28, with upregulated OCN expression. Therefore, the pharmacological effect of PLGA-NfDs can be considered twofold, as follows: (i) for PDL remodeling following OTM, reduced expression of cytokines such as TNF-α and IL-1β causes reduced fibroblast frequency, resulting in reduced RANKL levels and osteoclastogenesis; and (ii) for bone remodeling, reduced RANKL level caused by a reduction in the proportion of cytokine-induced osteoblasts also reduces RANKL/OPG, whereas increased TGF-β1 expression contributes to the pre-osteoblastic effect, leading to an increased osteoblast/osteocyte ratio. That is, periodontal health benefits and regeneration during OTM could be manifested if inflammatory reactions are effectively regulated.

In contrast to conventional periodontal socket healing methods during OTM [9,40,41,42], such as bone grafting, bone regeneration barrier membranes, and the application of platelet-rich fibrin, PLGA-NfDs offer a targeted genetic intervention approach on the NF-κB pathway underlying periodontal formation and resorption, possibly providing a rather precise therapeutic strategy. Regarding the mechanisms of action, conventional methods often provide physical or biochemical support to aid periodontal regeneration [43], whereas the PLGA-NfDs provide a targeted decoy ODN approach that essentially inhibits specific genetic pathways associated with periodontal resorption, concurrently promoting periodontal formation in OTM. However, when comparing the PLGA-NfDs to the established conventional periodontal regenerative treatments listed above, it is evident that while the nanospheres offer innovative advantages, they lack the long-standing clinical validation of conventional methods. Nonetheless, the PLGA-NfD strategy is non-invasive and avoids surgical complications and potential donor site morbidity that may compromise the treatment results [44,45]. While early indications are promising, further studies, including rigorous, long-term clinical trials and comparative studies against established methodologies, such as other contemporary regenerative periodontal surgery, along with addressing the concerns regarding optimal timing for OTM following tooth extraction, are imperative to gain a comprehensive understanding of the potential of PLGA-NfD in periodontal regeneration.

## 4. Materials and Methods

### 4.1. Decoy ODN Nuclear Medicine

In continuation of our previous research, naked scrambled ODNs (ScD), structured as phosphorothioated double-stranded ScD with sequences 5ʹ-TTGCCGTACCTGACTTAGCC-3′ and 3′-AACGGCATGGACTGAATCGG-5′, and NF-κB-targeting ODNs (NfD), structured as phosphorothioated double-stranded NfD with sequences 5′-CCTTGAAGGGATTTCCCTCC-3′ and 3′-GGAACTTCCCTAAAGGGAGG-5′, were used as previously described [17]. The concentrations of ScD and NfD in solutions containing ScD and NfD, as well as in PLGA-ScD and PLGA-NfD solutions, respectively. The concentration of hydroxypropyl cellulose-H with PLGA NS and the physicochemical properties of decoy ODN-loaded PLGA NS were determined according to Huang et al. [17]. Reagents for the NfD and PLGA-NfDs were provided by AnGes Inc. (Osaka, Japan) and Hosokawa Micron Corporation (Osaka, Japan).

### 4.2. Experimental Animals

Forty-six-week-old male Wistar/ST rats (Sankyo Lab Service, Tokyo, Japan) (Appendix A) were used in compliance with the ARRIVE 2.0 guidelines. All in vivo experiments were approved by the Institutional Animal Care and Use Committee of Tokyo Medical and Dental University (Approval No. A2022-116A). Ten rats were randomly assigned to four treatment groups: one negative control (ScD) or three experimental (NfD, PLGA-ScD, and PLGA-NfD) groups (Figure 1C). All rats were housed in the same room with controlled temperature, humidity, and light (standard alternating 12/12 h light/dark cycle). The health status and body weight of the rats were monitored every other day.

### 4.3. OTM in a Tooth-Extraction Animal Model

All rats were subjected to general anesthesia and surgery to bilaterally extract their maxillary first molar (M1) (Figure 1A). General anesthesia was induced by subcutaneous injection of a mixed anesthetic (0.3 mg/kg medetomidine [Nippon Zenyaku Kogyo Co., Ltd., Fukushima, Japan], 4 mg/kg midazolam [Teva Takeda Yakuhin Ltd., Nagoya, Japan], 5 mg/kg butorphanol [Meiji Seika Pharma Co., Ltd., Tokyo, Japan]). The bilateral M1 was extracted using forceps. After the surgery, specific nuclear medicines were administered through intraosseous injection (0.25 mL per extraction socket region). Subsequently, OTM was initiated using a 10-g force nickel–titanium closed coil tension spring (Tomy International, Tokyo, Japan) attached from the bilateral maxillary second molar (M2) to the upper incisors and fixed with light-cured composite resin (GC, Tokyo, Japan). Postoperative complications or syndromes were not observed in any of the rats.

### 4.4. In Vivo 3D Micro-CT Analysis

To obtain the linear and angular measurements of the OTM, 3D images of the maxilla were captured on post-extraction days (D)0, D7, and D28 using an in vivo 3D micro-CT scanner (R_mCT2 SPMD; Rigaku, Tokyo, Japan) (Figure 1B). The maxilla was scanned using 90-kV X-ray energy, 160-µA intensity, and a 30-mm field of view. Analysis software (TRI/3D-BON-FCS64; RATOC System Engineering Co., Ltd., Tokyo, Japan) was used to superimpose M2 OTM slice images over postoperative micro-CT data (D0) as baseline [46]. To measure tooth movement, we investigated the distance between the M2 distal contact point and the maxillary third molar mesial contact point. The tooth inclination angle was defined as the angle between the mesial root of the M2 and the occlusal plane (Appendix A).

### 4.5. Tissue Preparation

To evaluate the alveolar bone morphology and histomorphometry, a split-mouth design was used, where the left hemi-maxilla, along with the extraction socket and the surrounding tissue of M2, was used (*n* = 5). The right side was used for biochemical evaluation (*n* = 5), as reported by Huang et al. [17]. After active OTM on D7 and D28, five rats from each group were euthanized (Figure 1B). Maxillae with M1 tooth-extraction socket and the M2 surrounding tissue were immediately collected. The left hemi-maxilla was fixed with 4% paraformaldehyde (pH 7.4; Fujifilm Wako Pure Chemical Corp., Osaka, Japan) for 48 h at 4 °C for morphological and histomorphometric analyses. The samples were examined by micro-CT and then prepared for histologic examination by preparing paraffin-embedded sections following decalcification. Immediately after collection, the right hemi-maxilla was resected and transferred to liquid nitrogen for biochemical evaluation.

### 4.6. Ex Vivo 3D Micro-CT

Micro-CT scanning (inspeXio SMX-100CT; Shimadzu Corp., Kyoto, Japan) was performed on left hemi-maxillae samples at a high resolution (8-µm isotropic nominal resolution), with 75-kV energy and 140-µA intensity, according to the manufacturer’s instructions. For the analysis of 3D microstructural morphometry, the VOI was defined by the borders that included the total M1 socket region and the M2 surrounding periodontium. VOI1 had a grid area of 29.184 mm^3^ (LX: 2.4 mm, LY: 1.9 mm, LZ: 6.4 mm), whereas VOI2 covered the mesial region of M2 alveolus and had a grid area of 1.344 mm^3^ (LX: 2.4 mm, LY: 1.6 mm, LZ: 0.35 mm), as shown in Figure 2A. Trabecular bone parameters were analyzed using the bone volume fraction (BV/TV; %), BMD; mg/cm^3^), trabecular number (Tb.N; per mm), and trabecular separation (Tb.Sp; µm), which were determined using the direct-measures technique [47].

### 4.7. Histological Analysis

The images were analyzed using ImageJ software (version 1.52; National Institutes of Health, Bethesda, MD, USA). The region of interest (ROI) was determined as four images of the 330 × 409 μm region at 200× magnification on the mesial side of the M2 periodontal alveolus, which is considered representative of the M2 tooth movement modality (Appendix A). Analysis was performed after obtaining three randomized tissue sections for each sample with four specific images. A representative histological diagram of the ROI crestal side is shown for each staining experiment. The same investigator (I.S.) performed all measurements, and the mean value was used as the final measurement.

#### 4.7.1. TRAP and ALP Staining

Bone resorption and formation were assessed using a TRAP&ALP staining kit (Fujifilm Wako Pure Chemical Corp.) according to the manufacturer’s instructions. The number of multinucleated TRAP-positive cells and ALP-positive stained area (%) in the ROI were quantified and analyzed by a single examiner, and the average values were calculated.

#### 4.7.2. Immunohistochemistry

Sections were stained with the following primary antibodies for immunohistochemical analyses: anti-TNF-α (dilution ratio: 1:400; Bioss Inc., Woburn, MA, USA), anti-IL-1β (1:400; Bioss Inc.), anti-TGF-β1 (1:400; Bioss Inc.), anti-OPN (1:400; Abcam, Cambridge, UK), and anti-CTSK (1:400; Abcam). Briefly, after deparaffinization and rehydration, the sections were incubated with 3% hydrogen peroxide (Abcam) to quench the endogenous peroxidase activity; thereafter, the sections were blocked with specific animal serum for nonspecific binding and then incubated with specific primary antibodies and VECTASTAIN Elite ABC Rabbit IgG Kit (Vector Laboratories, Inc., Burlingame, CA, USA) with biotinylated secondary antibody and ABC Reagent. Subsequently, 3,3′-diaminobenzidine (Abcam) was applied, and hematoxylin (Fujifilm Wako Pure Chemical Corp.) staining was performed for counterstaining. The levels of TNF-α, IL-1β, TGF-β1, and OPN were semi-quantified by the percentage of immuno-positive stained areas (%).

#### 4.7.3. Immunofluorescence Analysis

Double immunofluorescent staining was performed using the following primary antibodies: mouse anti-NF-κB p65 (dilution ratio: 1:150; Santa Cruz Biotechnology, Dallas, TX, USA), rabbit anti-CTSK (1:150; Abcam), rabbit anti-RANKL (1:150; Proteintech Group Inc., Rosemont, IL, USA), rabbit anti-OPG (1:150; Abcam), rabbit anti-OPN (1:200; Abcam), and rabbit anti-OCN (1:200; Abcam). The sections were first blocked using 5% bovine serum albumin and were then incubated with a specific primary antibody, followed by species-matched secondary antibodies (Donkey Anti-Rabbit IgG H&L Alexa Fluor 594 and Donkey Anti-Rabbit IgG H&L Alexa Fluor 488; Abcam), at a 1:200 dilution. All tissue sections were subjected to autofluorescence quenching (Vector TrueVIEW; Vector Laboratories, Inc.) and mounted with VECTASHIELD Vibrance Antifade Mounting Medium with DAPI (Vector Laboratories, Inc.) following the manufacturer’s protocol. Fluorescent images were acquired using a Leica TCS-SP8 confocal laser scanning microscope (Leica Biosystems, Wetzlar, Germany) within 48 h after mounting. The immunofluorescence expression of each sample was evaluated using the MFI; a.u.).

### 4.8. Biochemical Evaluation

#### Quantitative Reverse Transcription-Polymerase Chain Reaction (RT-qPCR)

RT-qPCR was used to examine the expression of genes related to inflammation and bone metabolism. Total RNA was isolated from VOI1 using TRIzol reagent (Thermo Fisher Scientific Inc., Waltham, MA, USA), as described previously [17,48], followed by cDNA synthesis using PrimeScript RT Master Mix (Takara Bio Inc., Shiga, Japan), according to the manufacturers’ instructions. Real-time PCR analysis was performed using the Probe qPCR Mix (Takara Bio Inc.) and the Applied Biosystems 7500 Real-Time PCR System (Thermo Fisher Scientific Inc.). Appropriate specific TaqMan Gene Expression Assay primers (Thermo Fisher Scientific Inc.) were chosen for real-time PCR amplification of rat *Gapdh* (Rn01775763_g1), rat *Tnf* (Rn01525859_g1), rat *Il1b* (Rn00580432_m1), rat *Tnfsf11* (Rn00589289_m1), rat *Tnfrsf11b* (Rn00563499_m1), rat *Tgfb1* (Rn00572010_m1), rat *Spp1* (Rn00681031_m1), rat *Ctsk* (NM_031560.2), rat *Fgf2* (NM_019305.2), and rat *Bglap* (Rn00566386_g1). Relative gene expression was calculated using the comparative Ct method and normalized to *Gapdh* expression. To assess the capability and degree of bone resorption and turnover, we calculated the receptor activator of NF-κB ligand (RANKL)/osteoprotegerin (OPG) ratio by the relative expression of *Tnfsf11/Gapdh* over *Tnfrsf11b/Gapdh*.

### 4.9. Statistical Analysis

Normality was assessed using the Shapiro–Wilk test, and the equality of variances was evaluated using Levene’s test. For parametric analysis, intergroup comparisons were performed using the one-way analysis of variance, followed by Tukey’s post-hoc test (*n* = 5 for each group). Statistical analyses were performed using IBM SPSS Statistics for Windows (version 27.0; IBM Corp., Armonk, NY, USA) and GraphPad Prism 9 (version 9.3.1; GraphPad Software Inc., Boston, MA, USA). The results are presented as the mean ± standard deviation (*n* = 5 per group). Statistical significance was set at *p* < 0.05.

## 5. Conclusions

In conclusion, PLGA-NfDs can be considered a novel therapeutic agent for promoting OTM in clinical space closure cases, serving as potential regulators of bone remodeling during OTM.

## Figures and Tables

**Figure 1 ijms-25-05223-f001:**
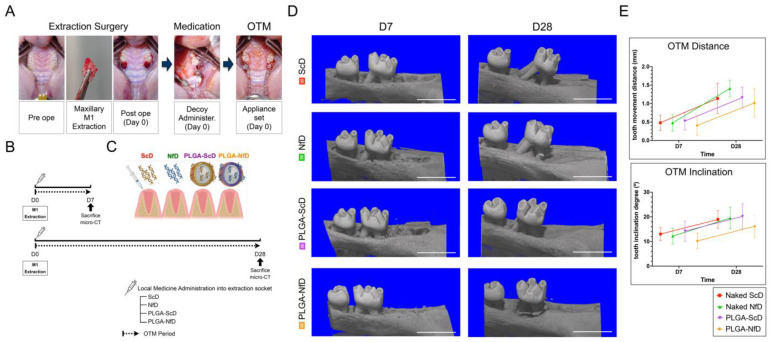
Orthodontic tooth movement (OTM) evaluation. (**A**) Intra-oral images illustrate the use of a 10-g force nickel–titanium coil spring in a sustained light-force OTM model. (**B**) Timeline of the study. (**C**) Schematic illustration of decoy oligodeoxynucleotide (ODN) administration in rats. (**D**) Representative 3D images of OTM on days 7 and 28 for each group (scale bar = 3 mm). (**E**) OTM quantification involving linear and angular measurements of the maxillary second molar (M2) using superimposition techniques to assess the impact of various orthodontic forces. Results show decreased OTM distance and inclination in rats treated with PLGA-NfD-loaded nanospheres, with no significant differences noted in other groups. D0—immediately after extraction; D7—post-extraction day 7; D28—post-extraction day 28; ScD—naked scrambled decoy group; NfD—naked nuclear factor-kappa B (NF-κB) decoy group; PLGA-ScD—scrambled decoy ODN-loaded PLGA nanosphere group; PLGA-NfD—NF-κB decoy ODN-loaded PLGA nanosphere group.

**Figure 2 ijms-25-05223-f002:**
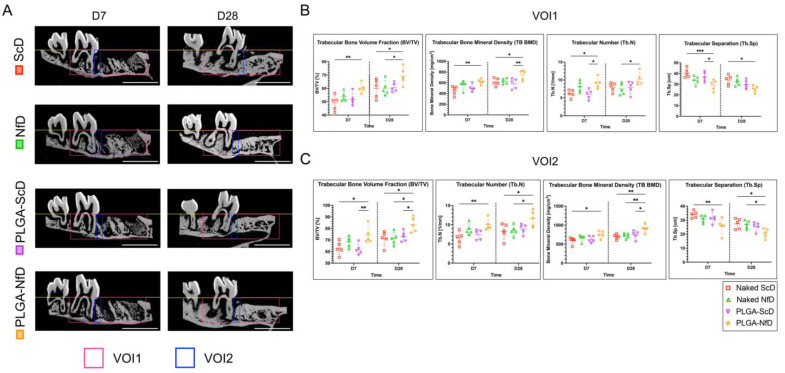
Micro-computed tomography (CT) analysis of trabecular bone. (**A**) Representative sagittal micro-CT images of alveolus changes: sagittal views show alveolus changes across groups. Pink lines define the volume of interest (VOI) for 3D microstructural morphometry around the primary maxillary molar (M1) and secondary molar (M2) periodontium (VOI1: 29.184 mm^3^; LX: 2.4 mm, LY: 1.9 mm, LZ: 6.4 mm). Blue lines mark the mesial region of the M2 alveolus (VOI2: 1.344 mm³; LX: 2.4 mm, LY: 1.6 mm, LZ: 0.35 mm; scale bar = 3 mm). (**B**,**C**) 3D micro-CT trabecular bone analysis of VOI1 and VOI2 on days 7 and 28 shows significant increases in bone volume (BV/TV), bone mineral density (BMD), and trabecular number (Tb.N) in the PLGA-NfD group, with reduced trabecular spacing (Tb.Sp) compared to that in the NfD and PLGA-ScD groups on both days. No significant differences between NfD and ScD effects were noted. Values are presented as mean ± standard deviation (SD; *n* = 5); * *p* < 0.05, ** *p* < 0.01, *** *p* < 0.001.

**Figure 3 ijms-25-05223-f003:**
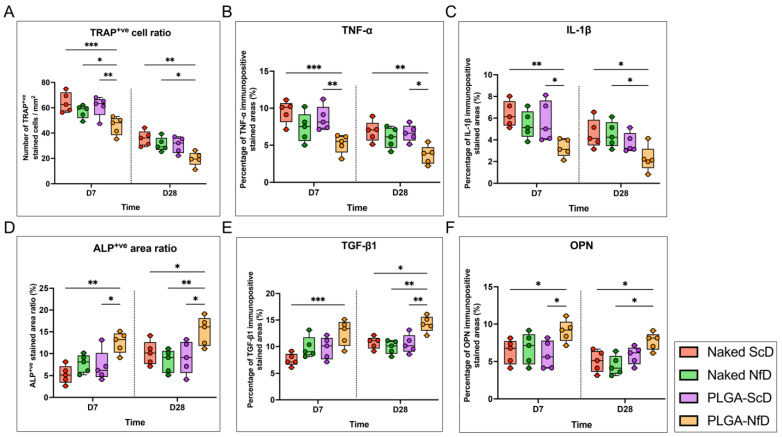
Representative findings of inflammatory, osteoclastogenic, and osteogenic histological assessments on D7 and D28 of orthodontic tooth movement (OTM) following extraction. Semi-quantitative analysis of (**A**) tartrate-resistant acid phosphatase (TRAP)-positive cells (per mm^2^) and (**D**) alkaline phosphatase (ALP)-positive area ratio (%). Percentage of immuno-positive stained areas (%) of (**B**) TNF-α, (**C**) IL-1β, (**E**) TGF-β1, and (**F**) OPN. The PLGA-NfD group showed fewer TRAP-positive osteoclasts per mm2 and a lower immuno-positive stained ratio of TNF-α to IL-1β than the other groups on D7 and D28. The PLGA-NfD group showed a significantly larger ALP-positive stained area and immuno-positive stained ratio for TGF-β1 and OPN than the other groups. Values are presented as mean ± standard deviation (SD; *n* = 5); * *p* < 0.05, ** *p* < 0.01, *** *p* < 0.001. D7—post-extraction day 7; D28—post-extraction day 28; NfD—naked NF-κB decoy group; PLGA-NfD—NF-κB decoy ODN-loaded PLGA nanosphere group; PLGA-ScD—scrambled decoy ODN-loaded PLGA nanosphere group; ScD—naked scrambled decoy group.

**Figure 4 ijms-25-05223-f004:**
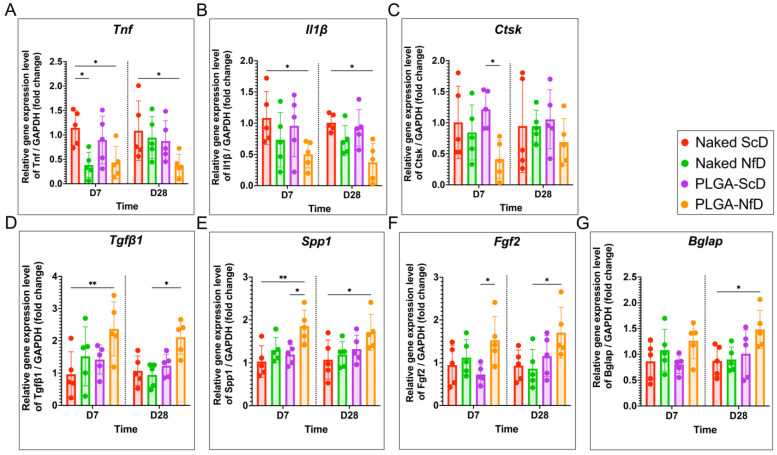
Representative findings of inflammatory, osteoclastogenic, and osteogenic biochemical assessments on D7 and D28 of orthodontic tooth movement (OTM) following extraction. Relative gene expression of inflammatory cytokines (**A**) *Tnf* and (**B**) *Il1b*, and osteoclastogenic marker (**C**) *Ctsk*. Relative gene expression of osteogenic markers (**D**) *Tgfb1*, (**E**) *Spp1*, and (**F**) *Bglap*, and growth factor (**G**) *Fgf2*. In the PLGA-NfD group, the expression of *Tnf* and *Il1b* decreased compared with that in other groups on D7 and D28, whereas the expression of *Ctsk* significantly decreased compared with that in the PLGA-ScD group on D7. On D7 and D28, the PLGA-NfD group showed significantly higher relative expression of *Tgfb1*, *Spp1*, *Fgf2*, and *Bglap*. Values are presented as mean ± standard deviation (SD; *n* = 5); * *p* < 0.05, ** *p* < 0.01. D7—post-extraction day 7; D28—post-extraction day 28; NfD—naked NF-κB decoy group; PLGA-NfD—NF-κB decoy ODN-loaded PLGA nanosphere group; PLGA-ScD—scrambled decoy ODN-loaded PLGA nanosphere group; ScD—naked scrambled decoy group.

**Figure 5 ijms-25-05223-f005:**
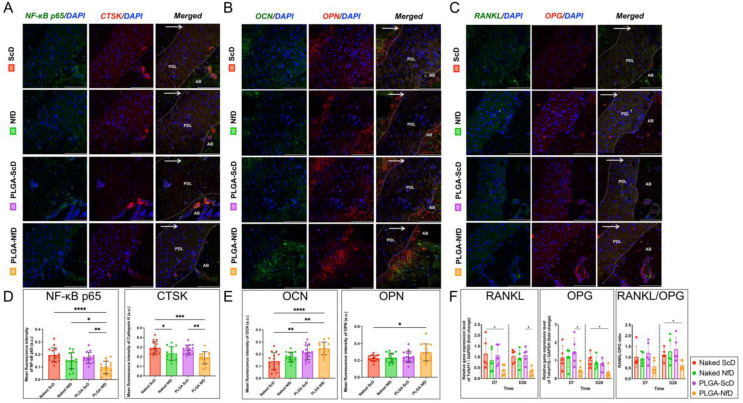
Representative findings of NF-κB p65 activity and osteoclastogenic and osteogenic histological and biochemical assessments on D7 and D28 of orthodontic tooth movement (OTM) following extraction. (**A**) Double immunofluorescent staining of NF-κB p65 and CTSK, (**B**) OCN and OPN, and (**C**) RANKL and OPG. (**D**,**E**) Semi-quantitative analysis of mean fluorescence intensity (MFI) of NF-κB p65, CTSK, OCN, and OPN. (**F**) Relative expression of *Tnfsf11* (RANKL) and *Tnfrsf11b* (OPG), and *Tnfsf11/Tnfrsf11b* (RANKL/OPG) ratio. On D28, the PLGA-NfD group showed less NF-κB p65 and CTSK MFI correspondingly, whereas OPN and OCN presented greater MFI. OCN did not show a similar enhanced immunofluorescent reaction compared with that in the other groups. The NfD group showed lower CTSK MFI values, and the PLGA-ScD group showed greater OCN MFI than the ScD group. On D7 and D28, the PLGA-NfD group showed a lower relative gene expression of RANKL and OPG, whereas decreased RANKL/OPG expression was only observed on D28. No significant difference was observed in the RANKL/OPG ratio between the groups on D7. A representative image (original magnification: 400×) is shown for each representative histological assay. White arrows and dotted lines indicate OTM direction and PDL borders, respectively. Scale bar = 100 μm. Values are presented as mean ± standard deviation (SD; *n* = 5); * *p* < 0.05, ** *p* < 0.01, *** *p* < 0.001, **** *p* < 0.0001. AB—alveolar bone; D7—post-extraction day 7; D28—post-extraction day 28; NfD—naked NF-κB decoy group; PDL—periodontal ligament; PLGA-NfD—NF-κB decoy ODN-loaded PLGA nanosphere group; PLGA-ScD—scrambled decoy ODN-loaded PLGA nanosphere group; ScD—naked scrambled decoy group.

## Data Availability

The original contributions presented in the study are included in the article/Appendix A, further inquiries can be directed to the corresponding author.

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
