# Peer review of "NF-κB Decoy Oligodeoxynucleotide-Loaded Poly Lactic-co-glycolic Acid Nanospheres Facilitate Socket Healing in Orthodontic Tooth Movement"

_ijms, 2024, doi:10.3390/ijms25105223_

Round 1

Reviewer 1 Report

Comments and Suggestions for Authors

Dear Authors,

Thank you for submitting the manuscript entitled 'NF-kB decoy oligodeoxynucleotide-loaded poly lactic-co-gly-colic acid nanospheres facilitate socket healing in orthodontic tooth movement`

The results indicated that PLGA-NfDs inhibited inflammation and induced osteogenesis in alveolar bone after tooth extraction during OTM. The authors clearly presented  the mechanism by using molecular biology and bonemorphometry. The design of the research was also done properly.  I beleive that the study will be an important basic knowledge to develop the  orthodontic treatment in the future.  I think it will be better if the authors make the discussion contents a little shorter and organize them.

Best regards

Author Response

Please see the attachment, thank you very much.

Reviewer 2 Report

Comments and Suggestions for Authors

The article is an animal model study on the delivery of NF-κB decoy oligodeoxynucleotide-loaded poly lactic-co-glycolic acid nanosphere on orthodontic extraction socket healing.

Abstract:

The abstract should contain the methods used.

Figures and legends

The figures and graphs are too small and blurred to allow for proper evaluation both in the main manuscript and supplemental file.

Once the revision is received, the manuscript can be evaluated more closely for scientific content and validity of the results. 

Round 2

Reviewer 2 Report

Comments and Suggestions for Authors

The authors have presented an animal study that looked at the use of NF-κB decoy oligodeoxynucleotide-loaded poly lactic-co-gly- 2 colic acid nanospheres in orthodontic tooth movement after tooth extraction. It is a well designed study. I would like to thank the authors for responding to comments and sending high resolution pictures and graphs in the letter. 

The work appears to be a follow up to this work "Huang AC, Ishida Y, Li K, Rintanalert D, Hatano-Sato K, Oishi S, Hosomichi J, Usumi-Fujita R, Yamaguchi H, Tsujimoto H, Sasai A, Ochi A, Watanabe H, Ono T. NF-κB Decoy ODN-Loaded Poly(Lactic-co-glycolic Acid) Nanospheres Inhibit Alveolar Ridge Resorption. Int J Mol Sci. 2023 Feb 12;24(4):3699."

One concern with the manuscript is the significant verbatim reproduction of material within the main text from other published sources especially from the paper cited above. 

In the abstract, the authors need to include more details such as sample size, method of study, ethical approval etc. Currently, the abstract goes from the aim to the results. Additionally, some points are focused and highlighted that are not really discussed in the manuscript, for example, "provoking periodontal regeneration" is highlighted in the abstract but not really discussed or focused on in the manuscript. I would urge the authors to rewrite the abstract per the journal guidelines and to include relevant content related to the methods. 

Comments on the Quality of English Language

Minor editing of English language. 

Author Response

Thank you for your kindest review. The response letter has been attached here and please kindly check. Thank you very much.
